# Narrowband Light Reflection Resonances from Waveguide Modes for High-Quality Sensors

**DOI:** 10.3390/nano10101966

**Published:** 2020-10-03

**Authors:** Ping Gu, Jing Chen, Chun Yang, Zhendong Yan, Chaojun Tang, Pinggen Cai, Fan Gao, Bo Yan, Zhengqi Liu, Zhong Huang

**Affiliations:** 1College of Electronic and Optical Engineering, Nanjing University of Posts and Telecommunications, Nanjing 210023, China; guping@njupt.edu.cn (P.G.); jchen@njupt.edu.cn (J.C.); 1018020724@njupt.edu.cn (C.Y.); 2College of Science, Nanjing Forestry University, Nanjing 210037, China; zdyan@njfu.edu.cn; 3Center for Optics and Optoelectronics Research, Collaborative Innovation Center for Information Technology in Biological and Medical Physics, College of Science, Zhejiang University of Technology, Hangzhou 310023, China; boyan@zjut.edu.cn; 4College of Physics Communication and Electronics, Jiangxi Normal University, Nanchang 330022, China; zliu@jxnu.edu.cn; 5College of Physics and Electronic Engineering, Jiangsu Second Normal University, Nanjing 210013, China; huangzhong89@126.com

**Keywords:** reflection resonances, narrow band, waveguide modes, sensors

## Abstract

Designing various nanostructures to achieve narrowband light reflection resonances is desirable for optical sensing applications. In this work, we theoretically demonstrate two narrowband light reflection resonances resulting from the excitations of the zero-order transverse magnetic (TM) and transverse electric (TE) waveguide modes, in a waveguide structure consisting of an Au sphere array on an indium tin oxide (ITO) spacer on a silica (SiO_2_) substrate. The positions of the light reflection resonances can be tuned easily, by varying the array periods of gold (Au) spheres or by changing the thickness of the ITO film. More importantly, the light reflection resonances have a very narrow bandwidth, the full width at half maximum (FWHM) of which can be reduced to only several nanometers for the zero-order TM and TE waveguide modes. The conventionally defined performance parameters of sensors, sensitivity (S) and figure of merit (FOM), have quite high values of about 80 nm/RIU and 32, respectively, in the visible wavelength range.

## 1. Introduction

Exploring different approaches to achieve high-quality refractive-index sensors has been drawing more and more of the attention of researchers in recent years, due to their great realistic significance and broad application prospect in the field of sensing detection. Among different approaches, surface plasmon resonances in a variety of metallic nanostructures have been studied extensively and intensively for refractive-index sensors [1,2,3,4,5,6,7,8,9,10,11,12,13,14,15,16]. Surface plasmon resonances are able to confine light into a sub-wavelength and nano-scale region, and simultaneously induce the great enhancement of electromagnetic fields in the vicinity of metallic nanostructures. When the refractive index of the environment medium around metallic nanostructures is changed, the spectral position and intensity of surface plasmon resonances will change, which is the fundamental physical mechanism that allows surface plasmon resonances to be applied for the detection of chemical and biological species [1,2]. In many cases, however, because of the process of electromagnetic radiation decay and the Joule heat produced by Ohmic loss in metal materials, surface plasmon resonances usually have a relatively broad bandwidth, and thus the electromagnetic field enhancement associated with surface plasmon resonances is limited, which is very detrimental to improving the sensing performances of refractive-index sensors [7,8]. In order to resolve such a problem, many methods for the sensitivity-enhancement of surface plasmon sensors have been theoretically proposed and experimentally demonstrated [9,10,11,12,13,14,15,16]. In the majority of cases, these methods depend on improving the electromagnetic fields of surface plasmon resonances and reducing the bandwidth of surface plasmon resonances as far as possible. In recent years, the plasmonic analogue of Fano resonance firstly discovered in atomic physics has been proven to be a very effective method to achieve ultra-sensitive refractive-index sensors [17,18,19,20,21,22,23,24,25,26,27,28,29]. In metallic nanostructures, the near-field interactions between a broadband, super-radiant (bright) plasmon mode and a narrowband, sub-radiant (dark) plasmon mode will lead to a Fano resonance with an anti-symmetric lineshape [18]. The plasmonic Fano resonance has an extremely narrow bandwidth associated with the great enhancement of electromagnetic fields, and thus is very sensitive to very small changes in the refractive index of the surrounding medium [19]. In a recent work [30], the interactions between waveguide modes and localized surface plasmon resonances in ellipsoidal Au nanoparticles are investigated theoretically to produce a sharp Fano resonance for refractive index sensors with good performance. The interaction condition is that the polarization direction of the localized surface plasmon resonances should match with the electric field direction of the waveguide modes. As a result, only TE waveguide modes can be utilized for such a sharp Fano resonance to appear. Recently, perfect absorbers of electromagnetic waves have also been proposed theoretically and demonstrated for the sensitivity-enhancement of refractive-index sensors [31,32,33,34,35,36,37,38,39,40,41,42,43,44,45,46]. The most significant advantage of this kind of refractive-index sensor is that it allows the detection of a refractive index change by an almost dark reference measurement, even if only a few photons are reflected from the perfect absorbers of electromagnetic waves [32]. If the perfect absorbers have a narrow bandwidth, extremely high sensing performances can be obtained for refractive-index sensors [33].

In this work, we theoretically study narrowband light reflection resonances in a waveguide structure consisting of an Au sphere array on an indium-tin oxide glass (ITO) spacer on a SiO_2_ substrate. It is found that the narrowband light reflection resonances result from the excitations of the zero-order TM and TE waveguide modes, by comparing the positions of the reflection resonances and the excitation wavelengths of the waveguide modes and by analyzing the distribution properties of the electromagnetic fields at the resonances. By varying the array periods of Au spheres, we can not only control what kind of waveguide modes (TM, TE or both of them) are excited, but also conveniently tune the positions of the reflection resonances. By changing the thickness of the ITO film, the positions of the reflection resonances can also be tuned easily. Moreover, the light reflection resonances have a very narrow bandwidth of only several nanometers, and their *FWHM* can be reduced to about 2.4 and 3.2 nm for the zero-order TM and TE waveguide modes, respectively. The numerical results show that two conventionally defined performance parameters related to sensors, *S* and *FOM*, have quite high values of about 80 nm/RIU and 32, respectively, in the visible wavelength range. In the current work, we mainly focus on narrowband light reflection resonances resulting from the excitations of both TE and TM waveguide modes, not considering Fano-like resonances due to the interactions between waveguide modes and localized surface plasmon resonances [30]. The most important role played by the Au nanospheres is adding the momentum of incident light by a reciprocal vector to excite the waveguide modes. An obvious difference between the current work and Ref. 30 is that TM waveguide modes can be excited efficiently. Furthermore, with the help of waveguide theory, we can satisfactorily predict the positions of the narrowband light reflection resonances.

## 2. Methods

The periodic array of Au spheres, with a diameter of *d* and different complex permittivities that dependent on the light wavelength [47], is put on the top surface of an ITO film with a thickness of *t* and a relative permittivity of 3.8, which is supported by a SiO_2_ substrate with a relative permittivity of 2.1 (see Figure 1). The above structure is able to excite the TE or TM waveguide modes that are highly confined and propagating in the ITO film, which cause narrowband light reflection resonances for sensing applications. The most important role played by the sphere array is the adding of the momentum of the incident light by a two-dimensional reciprocal vector in order to match the momentum of waveguide modes (that is, the conservation law of momentum) [48]. By changing the array period (*p_x_* and *p_y_*) or the ITO thickness (*t*), the resonance wavelength of the waveguide modes can be tuned conveniently. The light is normally incident from above to below, and is polarized in the direction of the *x* axis. The commercial software package “EastFDTD” [49,50,51] is utilized for calculating the reflection spectra and the electrical and magnetic fields at reflection resonances.

## 3. Results and Discussion

For the array periods *p_x_* = *p_y_* = 400 nm, there are no obvious spectral features to be observed in Figure 2a, since no other waveguide modes are excited for such a short period. However, when the array period *p_y_* is increased to 500 nm in Figure 2b, we see a strong reflection peak at the resonance wavelength of *λ_1_* = 793 nm, due to the excitation of a TE waveguide mode. Similarly, for the array period *p_x_* to be increased to 500 nm in Figure 2c, another strong reflection peak will appear at the resonance wavelength of *λ_2_* = 739 nm, which is owing to the excitation of a TM waveguide mode. Very interestingly, when both *p_x_* and *p_y_* are increased to 500 nm in Figure 2d, we observe two reflection peaks at the resonance wavelengths of *λ_1_* and *λ_2_*, because the TE and TM waveguide modes are simultaneously excited in this case. As is clearly seen in Figure 2, we can easily control what kind of waveguide modes are excited by varying the array periods of Au spheres. In the following sections, we will demonstrate that the above reflection peaks are indeed because of the excitations of different kinds of waveguide modes.

To prove that the reflection peaks at the resonance wavelengths of *λ_1_* and *λ_2_* are caused respectively by the TE and TM waveguide modes, we have plotted the corresponding electromagnetic field distributions in Figure 3. For the array periods *p_x_* = 400 nm and *p_y_* = 500 nm, the period *p_x_* is too short to open a propagation channel in the *x* direction. Therefore, only a waveguide mode is excited, which can only propagate in the *y* direction. This point can be verified via the characteristic distributions of alternating electronic and magnetic fields along the propagation direction (i.e., the *y* direction), as shown in Figure 3a,b. The red arrows in Figure 3a clearly indicate that the waveguide mode has an electric field perpendicular to the propagation direction, which is thus called the TE waveguide mode [52]. In contrast, for the array periods *p_x_* = 500 nm and *p_y_* = 400 nm, the propagation channel of waveguide modes is closed in the *y* direction, but is opened in the *x* direction. So, it is seen in Figure 3c,d that the electronic and magnetic fields alternately present along the *x* direction. The magnetic field of the waveguide mode is perpendicular to the propagation direction, as indicated by the red arrows in Figure 3d. In this situation, such a waveguide mode is usually defined as the TM waveguide mode [52].

By changing the array periods of Au spheres, we can not only control which type of waveguide mode is excited, but also can tune the resonance wavelengths of the waveguide modes to shift the positions of the reflection peaks. For the array periods *p_x_* and *p_y_* to increase from 460 to 540 nm in steps of 20 nm in Figure 4a–e, two reflection peaks move obviously to the long-wave band, because the resonance wavelengths of the waveguide modes are red-shifted. With the increasing periods, the bandwidths of two reflection peaks become narrower and narrower, which is attributed to the much lower radiation damping of waveguide modes [48]. The positions of two reflection peaks for different periods are given in Figure 4f, which are in good agreement with the resonance wavelengths of the zero-order TE and TM waveguide modes. This further confirms that the reflection peaks are really related to the excitations of waveguide modes. The resonance wavelengths of zero-order waveguide modes are calculated by the following equations [52]:2*kn*_1_*t*cos*θ*_1_ − 2Φ_10_ − 2Φ_12_ = 0,*θ*_1_ > sin^−1^(*n*_0_/*n*_1_),*β* = *kn*_1_sin*θ*_1_ = *G*,(1)

For TE-waveguide mode, they are
tanΦ_10_ = √(*n*_1_^2^sin^2^*θ*_1_ − *n*_2_^2^)/(*n*_1_cos*θ*_1_)tanΦ_12_ = √(*n*_1_^2^sin^2^*θ*_1_ − *n*_0_^2^)/(*n*_1_cos*θ*_1_)(2)
and for TM-waveguide mode, they are
tanΦ_10_ = √(*n*_1_^2^sin^2^*θ*_1_ – *n*_2_^2^)/(*n*_2_^2^*n*_1_cos*θ*_1_)tanΦ_12_ = √(*n*_1_^2^sin^2^*θ*_1_ – *n*_0_^2^)/(*n*_0_^2^*n*_1_cos*θ*_1_)(3)
where *n*_2_, *n*_1_ and *n*_0_ are the refractive indices of air, ITO and SiO_2_, respectively. *θ_1_* is the angle between the inner normal of the ITO interface and the light path. *k* is the wave vector of light in free space, and *G* is the reciprocal vector of the two-dimensional sphere array.

By varying the thickness *t* of the ITO film, the resonance wavelengths of the waveguide modes can also be tuned to shift the positions of the reflection peaks. When the ITO thickness *t* is increased from 120 to 200 nm in steps of 20 nm, the reflection peaks will move quickly to the long-wave band, due to the red-shift of the zero-order TE and TM waveguide modes, as shown in Figure 5a–e. The resonance wavelengths of the waveguide modes predicted by the above equations are again in an excellent agreement with the positions of the reflection peaks, as exhibited in Figure 5f. This fully testifies to the correlations between the reflection peaks [52].

To investigate the potential applications of high-quality sensors, which are based on the narrowband light reflection resonances from waveguide modes, in Figure 6a–d we calculate the reflection spectra for the varied refractive index *n* of the environment medium surrounding the Au spheres. As the refractive index *n* varies from 1.0 to 1.3 in steps of 0.1, two reflection peaks from the TM and TE waveguide modes red-shift significantly, which move from 775.2 and 831.9 nm to 798.7 and 855.7 nm, as presented in Figure 6e. So, we can achieve a conventionally defined sensitivity [31,53,54,55,56,57], *S* = Δ*λ*/Δ*n* = 78.3 and 79.3 nm/RIU, respectively. For the refractive index *n* = 1.0, the full width at half maximum *FWHM* of two reflection peaks is 2.4 and 3.2 nm, respectively. Then, the conventionally defined figure of merit [31,53,54,55,56,57] *FOM* = *S*/*FWHM* = 32.6 and 24.8, respectively. *FOM* can not only indicate the sensitivity of the reflection peaks to the change of *n*, but can also indicate the bandwidths of reflection peaks. Decreasing *FWHM* is an effective approach to improving *FOM*, and more importantly a small *FWHM* is also greatly beneficial to measuring the intensity change of reflected light in some sensing applications [31,32,33]. In our work, the *FWHM* of two reflection peaks can be reduced to only several nanometers, although it will be broadened when *n* is increased. For practical applications, *FWHM* is usually achieved with the reflection or transmission spectra for *n* = 1.0 [33]. The above two performance parameters related to sensors are quite high values in the visible wavelength range, with respect to those of the refractive index sensors based on localized surface plasmon resonances [2].

## 4. Conclusions

In summary, we have theoretically studied narrowband light reflection resonances in a waveguide structure consisting of an Au sphere array on an ITO spacer on a SiO_2_ substrate. It is found that the narrowband light reflection resonances result from the excitations of the zero-order TM and TE waveguide modes, by comparing the positions of the reflection resonances and the excitation wavelengths of the waveguide modes and by analyzing the distribution properties of electromagnetic fields at resonances. By varying the array periods of Au spheres, we can not only control what kind of waveguide modes (TM, TE or both of them) are excited, but also conveniently tune the positions of the reflection resonances. By changing the thickness of the ITO film, the positions of the reflection resonances can also be tuned easily. Moreover, the light reflection resonances have a very narrow bandwidth of only several nanometers, and their *FWHM* can be reduced to about 2.4 and 3.2 nm for the zero-order TM and TE waveguide modes, respectively. The numerical results show that the two conventionally defined performance parameters related to sensors, *S* and *FOM*, have quite high values of about 80 nm/RIU and 32, respectively, in the visible wavelength range.

## Figures and Tables

**Figure 1 nanomaterials-10-01966-f001:**
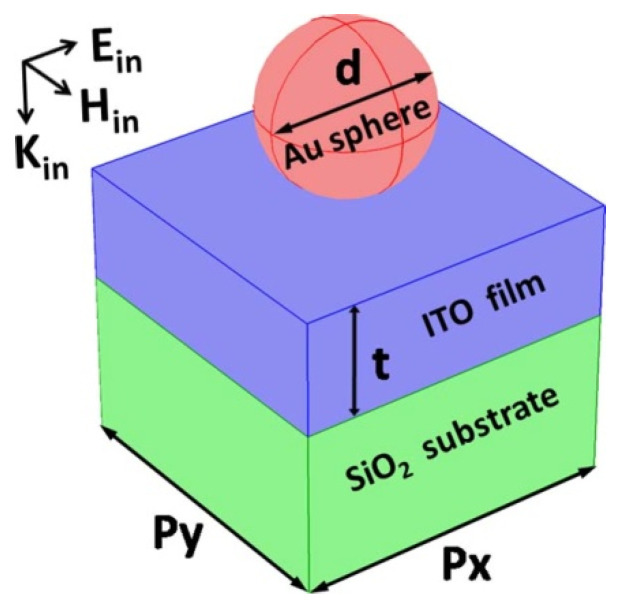
Schematic of a high-quality sensor based on narrowband light reflection resonances from waveguide modes.

**Figure 2 nanomaterials-10-01966-f002:**
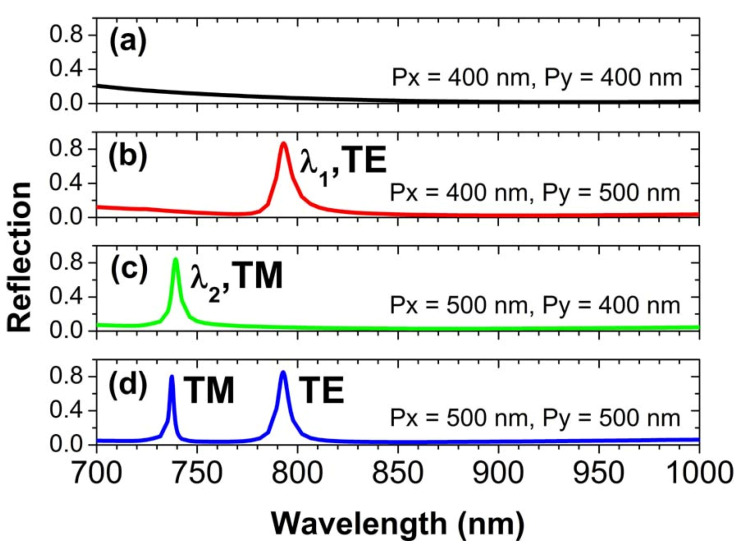
The reflection spectra of normally incident light for the array periods of Au spheres to be changed. (**a**) For the array periods *p_x_* = 400 nm and *p_y_* = 400 nm; (**b**) For the array periods *p_x_* = 400 nm and *p_y_* = 500 nm; (**c**) For the array periods *p_x_* = 500 nm and *p_y_* = 400 nm; (**d**) For the array periods *p_x_* = 500 nm and *p_y_* = 500 nm.

**Figure 3 nanomaterials-10-01966-f003:**
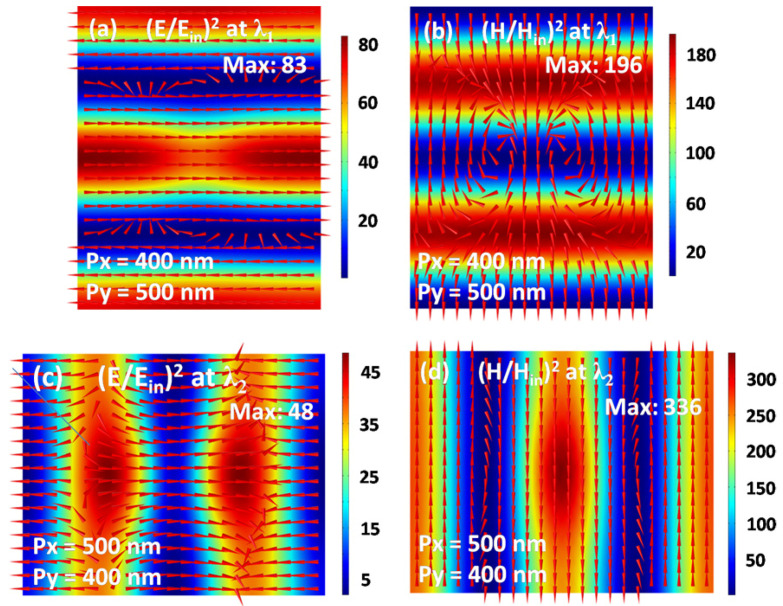
The intensity distributions on the *xoy* plane of electromagnetic fields in the ITO film for the resonance wavelength λ_1_ (**a,b**) and the resonance wavelength λ_2_ (**c,d**).

**Figure 4 nanomaterials-10-01966-f004:**
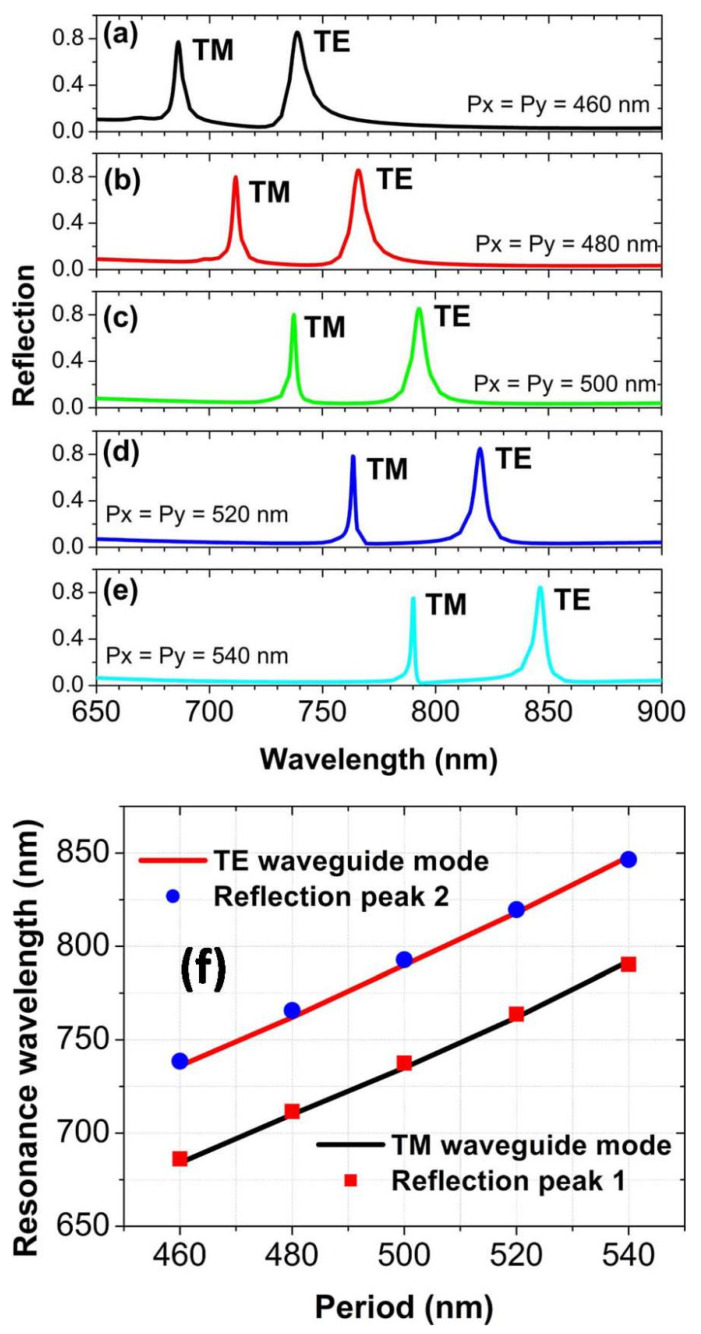
(**a**–**e**) The reflection spectra at normal incidence for different array periods of Au spheres. (**f**) Compare the positions of reflection peaks with the resonance wavelengths of waveguide modes.

**Figure 5 nanomaterials-10-01966-f005:**
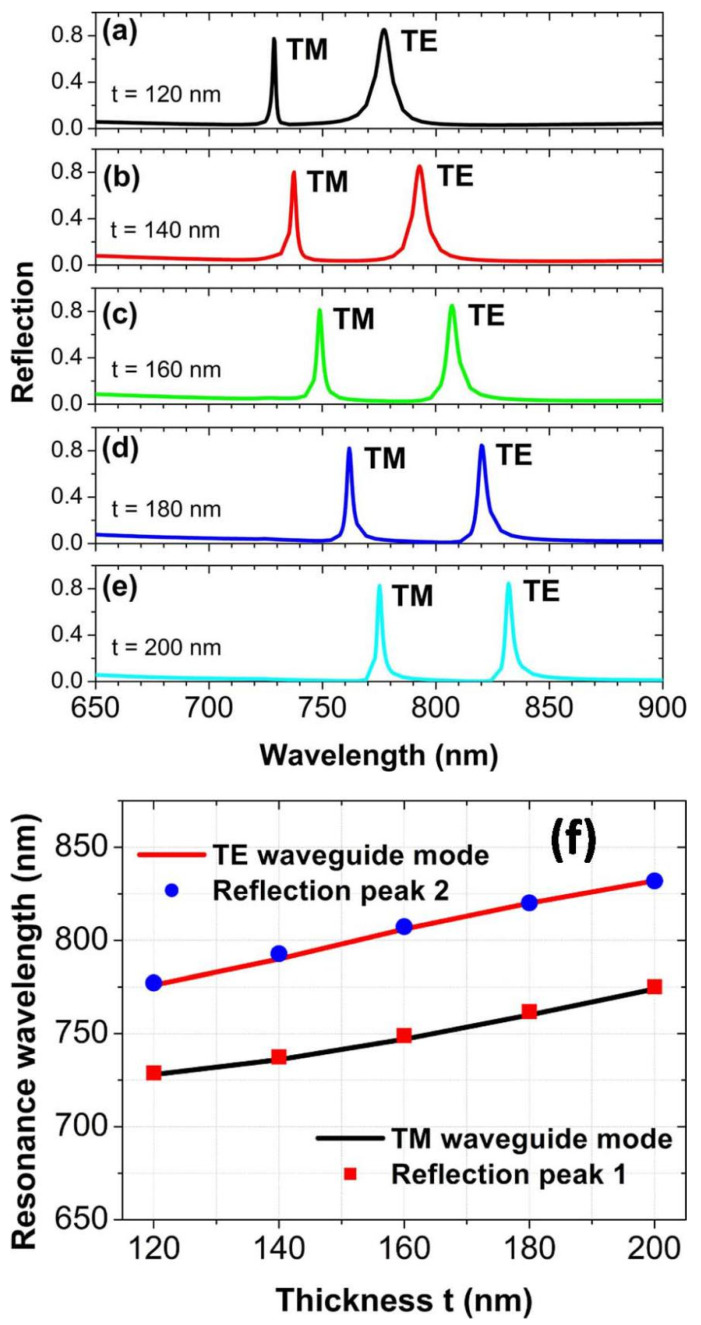
(**a**–**e**) The reflection spectra at normal incidence for different thicknesses of ITO film. (**f**) Compare the positions of reflection peaks with the resonance wavelengths of waveguide modes.

**Figure 6 nanomaterials-10-01966-f006:**
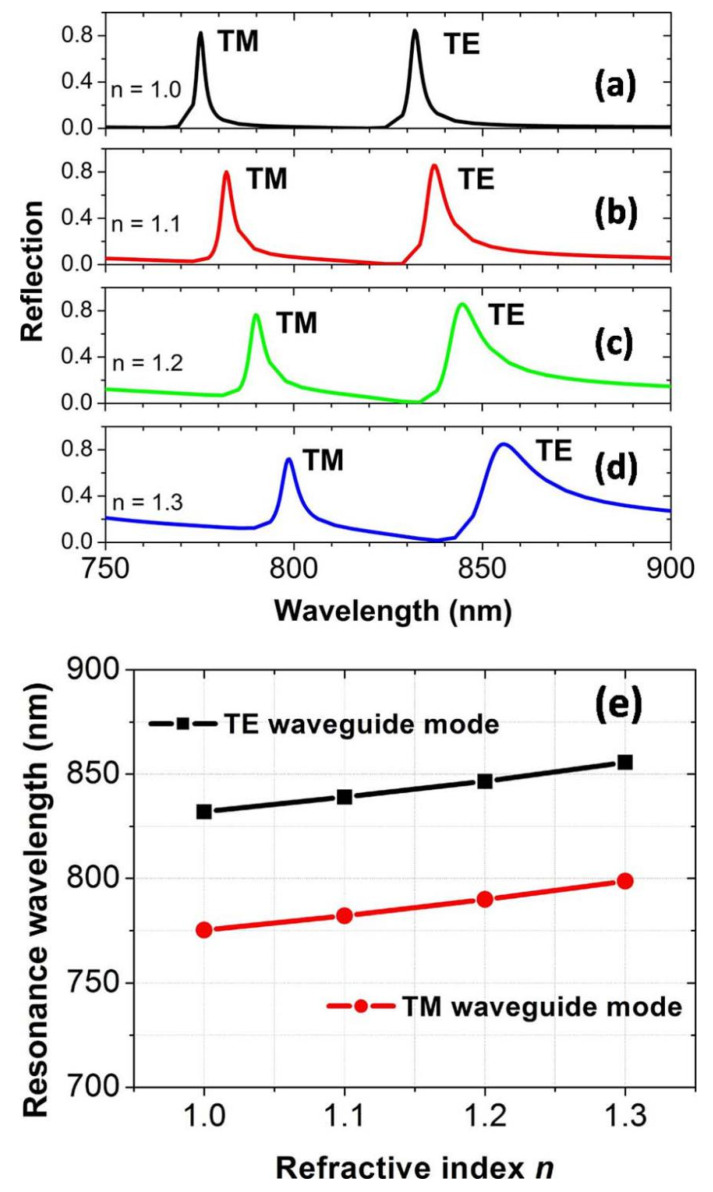
(**a**–**d**) The reflection spectra at normal incidence for a different refractive index. (**e**) The dependence of the reflection peak on the refractive index.

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
