# Peer review of "Narrowband Light Reflection Resonances from Waveguide Modes for High-Quality Sensors"

_nanomaterials, 2020, doi:10.3390/nano10101966_

Round 1

Reviewer 1 Report

This is an interesting theoretical work. I have just few minor comments and suggestions:

  1. In the Abstract and Conclusion sections, I suggest stating the values of the performance parameters, S and FOM. Thus, you may rewrite the last parts of Line 26 and Line 184 as: "have quite high values of about 80 nm/RIU and 32, respectively, in the visible wavelength range."

  2. Write all acronyms in full the first time they are used in the Abstract and in the Body of the paper. For example, in Line 20 and Line 63, if ITO means indium tin oxide, then you should write Indium Tin Oxide (ITO).

  3. Line 17: Delete "will". That is, “In this work, we will theoretically” should be corrected to “In this work, we theoretically”.

  4. If possible, you should improve the quality of Figure 1. The lines are not smooth; but this may be due the software used for the drawing.

Author Response

This is an interesting theoretical work. I have just few minor comments and suggestions: 1. Comment: In the Abstract and Conclusion sections, I suggest stating the values of the performance parameters, S and FOM. Thus, you may rewrite the last parts of Line 26 and Line 184 as: "have quite high values of about 80 nm/RIU and 32, respectively, in the visible wavelength range." Response: According to your helpful suggestion, we have rewritten the relevant sentences. Please see Line 27 and Line 206. 2. Comment: Write all acronyms in full the first time they are used in the Abstract and in the Body of the paper. For example, in Line 20 and Line 63, if ITO means indium tin oxide, then you should write Indium Tin Oxide (ITO). Response: According to your valuable suggestion, we have defined all abbreviations the first time cited in the text. Please see the part of abstract. 3. Comment: Line 17: Delete "will". That is, “In this work, we will theoretically” should be corrected to “In this work, we theoretically”. Response: According to your good suggestion, we have deleted the word “will”. Please see Line 17. 4. Comment: If possible, you should improve the quality of Figure 1. The lines are not smooth; but this may be due the software used for the drawing. Response: According to your valuable suggestion, we have improved the quality of this figure. Please see Figure 1.

Reviewer 2 Report

The article reports that a waveguide structure consisting of an Au sphere array on ITO/SiO2 produces to 2 ultra-narrowband light reflection resonances when the zero-order TM and TE waveguide modes are excited, which may be used for sensing applications (with high-value sensitivity and figure of merit in the visible wavelength range). The paper is very well written, and the figures are very clear. The results are interesting.

Author Response

We thank the referee very much for her/his valuable comments that help us to improve the quality of our paper. We have addressed all the questions one-by-one.

1. Comment: The article reports that a waveguide structure consisting of an Au sphere array on ITO/SiO2 produces to 2 ultra-narrowband light reflection resonances when the zero-order TM and TE waveguide modes are excited, which may be used for sensing applications (with high-value sensitivity and figure of merit in the visible wavelength range). The paper is very well written, and the figures are very clear. The results are interesting.

Response: We greatly thank the referee for her/his positive comments on our work.

Reviewer 3 Report

The use of a 2D array of gold nanoparticle on top of a dielectric waveguide to enhance localised surface plasmon resonance (LSPR) sensing has already been demonstrated experimentally (see e.g. Chen, J., Yuan, J., Zhang, Q., Ge, H., Tang, C., Liu, Y., & Guo, B. (2018). Dielectric waveguide-enhanced localized surface plasmon resonance refractive index sensing. Optical Materials Express, 8(2), 342. https://doi.org/10.1364/ome.8.000342). Therefore, the authors need to rewrite their introduction to clarify novelty, as well as clearly describe their contribution to the field. 

The Q-factor of the observed resonances are on the order of 100. The resonances are narrowband compared to LSPR, but it is misleading to use the term "ultra-narrowband". The term "ultra-narrowband" should therefore be replaced by "narrowband". 

Author Response

1.Comment: The use of a 2D array of gold nanoparticle on top of a dielectric waveguide to enhance localised surface plasmon resonance (LSPR) sensing has already been demonstrated experimentally (see e.g. Chen, J., Yuan, J., Zhang, Q., Ge, H., Tang, C., Liu, Y., & Guo, B. (2018). Dielectric waveguide-enhanced localized surface plasmon resonance refractive index sensing. Optical Materials Express, 8(2), 342. https://doi.org/10.1364/ome.8.000342). Therefore, the authors need to rewrite their introduction to clarify novelty, as well as clearly describe their contribution to the field.

Response: According to your valuable suggestion, we have discussed the differences between the paper mentioned above and our present work, and also clarified the novelty of this work, as the following: “In a recent work [57], the interactions between waveguide modes and localized surface plasmon resonances in ellipsoidal Au nanoparticles are investigated theoretically to produce a sharp Fano resonance for refractive index sensors with good performance. The interaction condition is that the polarization direction of localized surface plasmon resonances should match with the electric field direction of waveguide modes. As a result, only TE waveguide modes can be utilized for such a sharp Fano resonance to appear. In this work, we mainly focus on narrowband light reflection resonances resulting from the excitations of both TE and TM waveguide modes. The most important role played by the Au nanospheres is adding the momentum of incident light by a reciprocal vector to excite the waveguide modes. So, the Au nanospheres can be replaced by other periodic nanostructures, including dielectric nanoparticle array that does not support surface plasmon resonances. Furthermore, with the help of waveguide theory [51], we can satisfactorily predict the positions of the narrowband light reflection resonances.” Please see the paragraph from Line 180 to Line 192.

2.Comment: The Q-factor of the observed resonances are on the order of 100. The resonances are narrowband compared to LSPR, but it is misleading to use the term "ultra-narrowband". The term "ultra-narrowband" should therefore be replaced by "narrowband".

Response: According to your helpful suggestion, we have replaced the term “ultra-narrowband” with the term “narrowband”. Please see Line 2, Line 17, Line 19, Line 64, Line 66, Line 83, Line 92, Line 160, Line 194, and Line 196.

Reviewer 4 Report

Review for the manuscript:

Entitled: "Ultra-narrowband Light Reflection Resonances from Waveguide Modes for High-Quality Sensors"

for Nanomaterials.

With ID: nanomaterials-942400

Dear authors,

Thank you for your manuscript.

General comments

Comments for the Authors

This work is well within the scope of Nanomaterials and it may be of interest to most of the readers of this journal. It is well organized with good references to follow. The main concern is that the presented FOM is not justified. More effort should be given for the discussion of this issue. Furthermore, Turnitin showed increased text overlap with previous published studies. Some issues should be addressed as listed in the specific comments below. For all the above I have opted to recommend a Major Revision for the current form of the manuscript.

Specific comments

P1, L20: ‘Au’,ITO’, ‘SiO2’ etc. Please define all abbreviations the first time cited in the text.

P3, Figs.2,4,5,6: Please correct ‘Refection’.

P8, L165: ‘figure of merity’. Please correct.

P8, L166: ‘The above two performance parameters about sensors are quite high values’ With respect to what? Please revise.

Author Response

1.Comment: This work is well within the scope of Nanomaterials and it may be of interest to most of the readers of this journal. It is well organized with good references to follow. The main concern is that the presented FOM is not justified. More effort should be given for the discussion of this issue.Furthermore, Turnitin showed increased text overlap with previous published studies. Some issues should be addressed as listed in the specific comments below. For all the above I have opted to recommend a Major Revision for the current form of the manuscript.

Response: According to your valuable suggestion, we have carefully checked the definition of FOM, which is correct. Please see equation (3) in reference 32. We have also further discussed FOM, as the following: “FOM not only can indicate the sensitivity of reflection peaks to the change of n, but also can indicate the bandwidths of reflection peaks. Decreasing FWHM is an effective approach to improve FOM, and more importantly small FWHM is also greatly beneficial to measuring the intensity change of reflected light in some sensing applications [30-32]. In our work, the FWHM of two refection peaks can be reduced to only several nanometers, although it will be broadened when n is increased. For practical applications, FWHM is usually achieved from the reflection or transmission spectra for n = 1.0 [32].” Please see this paragraph from Line 168 to Line 174.

2.Comment: P1, L20: ‘Au’, ‘ITO’, ‘SiO2’ etc. Please define all abbreviations the first time cited in the text.

Response: According to your valuable suggestion, we have defined all abbreviations the first time cited in the text. Please see the part of abstract.

3.Comment: P3, Figs.2,4,5,6: Please correct ‘Refection’.

Response: We greatly thank you for your pointing out this spelling error. We have corrected it. Please see Figs.2, 4, 5 and 6.

4.Comment: P8, L165: ‘figure of merity’. Please correct.

Response: Thank you very much for your pointing out the spelling error. We have corrected it. Please see Line 167.

5.Comment: P8, L166:‘The above two performance parameters about sensors are quite high values’ With respect to what? Please revise.

Response: According to your helpful suggestion, we have revised this sentence, as the following: “The above two performance parameters about sensors are quite high values in the visible wavelength range, with respect to those of refractive index sensors based on localized surface plasmon resonances [2].” Please see Line 175 and Line 176.

Round 2

Reviewer 3 Report

The new paragraph (lines 180-192) discussing the earlier work by Chen et. al. [57] on interaction between LSPR and waveguide modes, now placed at the end of section 3 (Results and discussion), should be moved (at least part of it) to section 1 (Introduction) where other background material is presented. The description should also be modified to account for the fact that or TE mode excitation, the only difference between the current work and [57] is that the latter measured in transmission rather than reflection. The physical mechanism is the same. Therefore, the current work should be presented as an extension of the work by Chen et.al. The most important new finding is that also TM modes can be excited. 

The sentence (lines 188-190) "So, the Au nanospheres can be replaced by other periodic nanostructures, including dielectric nanoparticle array that does not support surface plasmon resonances." describes a conventional grating coupler (with complex grating structure) and as such is no novelty. Either the sentence should be removed or a reference to grating coupling should be added.

Author Response

We thank the referee very much for her/his valuable comments that help us to improve the quality of our paper. We have addressed all the questions one-by-one.

1.Comment: The new paragraph (lines 180-192) discussing the earlier work by Chen et. al. [57] on interaction between LSPR and waveguide modes, now placed at the end of section 3 (Results and discussion), should be moved (at least part of it) to section 1 (Introduction) where other background material is presented.The description should also be modified to account for the fact that or TE mode excitation, the only difference between the current work and [57] is that the latter measured in transmission rather than reflection. The physical mechanism is the same. Therefore, the current work should be presented as an extension of the work by Chen et.al. The most important new finding is that also TM modes can be excited.

Response: According to your helpful suggestion, the discussions on the earlier work are now placed into the part of introduction. Please see the paragraph from Line 57 to Line 62. We have further clarified the differences between the current work and the previous paper mentioned above, and also modified some relevant descriptions of the differences. Please see the paragraph from Line 82 to Line 89.

2.Comment: The sentence (lines 188-190) "So, the Au nanospheres can be replaced by other periodic nanostructures, including dielectric nanoparticle array that does not support surface plasmon resonances." describes a conventional grating coupler (with complex grating structure) and as such is no novelty. Either the sentence should be removed or a reference to grating coupling should be added.

Response: According to your valuable suggestion, we have removed the mentioned sentence that is not able to show the novelty of this work. Please see the paragraph from Line 82 to Line 89.

Reviewer 4 Report

Review for the manuscript:

Entitled: "Ultra-narrowband Light Reflection Resonances from Waveguide Modes for High-Quality Sensors"

for Nanomaterials.

With ID: nanomaterials-942400.R1

Dear authors,

Thank you for your manuscript.

General comments

Comments for the Authors

Authors responded to all reviewers’ remarks. Now I feel that the manuscript is publishable.

Best regards

Author Response

We thank the referee very much for her/his valuable comments that help us to improve the quality of our paper. We have addressed all the questions one-by-one.

1.Comment: Authors responded to all reviewers’ remarks. Now I feel that the manuscript is publishable.

Response: We greatly thank the referee for her/his positive comments on our responses.